# Exploring the Therapeutic Potential of Targeting GH and IGF-1 in the Management of Obesity: Insights from the Interplay between These Hormones and Metabolism

**DOI:** 10.3390/ijms24119556

**Published:** 2023-05-31

**Authors:** Sarmed Al-Samerria, Sally Radovick

**Affiliations:** Laboratory of Human Growth and Reproductive Development, Department of Pediatrics, Robert Wood Johnson Medical School, Rutgers, The State University of New Jersey, New Brunswick, NJ 08901, USA; sr1123@rwjms.rutgers.edu

**Keywords:** obesity, growth hormone (gh), insulin-like growth factor 1 (igf-1), metabolism, body composition, lipolysis, energy expenditure

## Abstract

Obesity is a growing public health problem worldwide, and GH and IGF-1 have been studied as potential therapeutic targets for managing this condition. This review article aims to provide a comprehensive view of the interplay between GH and IGF-1 and metabolism within the context of obesity. We conducted a systematic review of the literature that was published from 1993 to 2023, using MEDLINE, Embase, and Cochrane databases. We included studies that investigated the effects of GH and IGF-1 on adipose tissue metabolism, energy balance, and weight regulation in humans and animals. Our review highlights the physiological functions of GH and IGF-1 in adipose tissue metabolism, including lipolysis and adipogenesis. We also discuss the potential mechanisms underlying the effects of these hormones on energy balance, such as their influence on insulin sensitivity and appetite regulation. Additionally, we summarize the current evidence regarding the efficacy and safety of GH and IGF-1 as therapeutic targets for managing obesity, including in pharmacological interventions and hormone replacement therapy. Finally, we address the challenges and limitations of targeting GH and IGF-1 in obesity management.

## 1. Introduction

Obesity is a prevalent health condition associated with an increased risk of developing several chronic illnesses, including dyslipidemia, type 2 diabetes mellitus (T2DM), hypertension, cardiovascular disease, and certain types of cancer [1]. According to the World Health Organization (WHO), in 2016, over 1.9 billion adults (18 years and older) were overweight, with more than 650 million being classified as obese. This represents a significant increase in the prevalence of obesity, more than doubling since 1980. The Centers for Disease Control and Prevention (CDC) report that 42.4% of adults in the United States were classified as obese in 2020. This trend is particularly pronounced among middle-aged, Hispanic, and non-Hispanic black adults. Moreover, childhood and adolescent obesity is a significant public health issue in the United States, with nearly one in five children/adolescents aged two to nineteen years being categorized as obese.

Obesity is a complex condition caused by a combination of genetic, environmental, and behavioral factors, including diet, physical activity, and exposure to endocrine-disrupting chemicals. It is characterized by an excess accumulation of body fat resulting from an ongoing positive energy balance (a higher intake of calories than expenditure) and insufficient physical activity, which disrupts the energy balance and normal physiological homeostasis [2]. The fundamental components of energy balance include energy intake, energy expenditure, and energy storage, which interact in a complex manner to affect body weight and the fat ratio [3,4]. The management of obesity is best approached through a combination of lifestyle modifications and physical activity. This involves the implementation of a balanced and nutritious diet, reduced sedentary behaviors, and regular exercise. In recognition of the demanding nature of modern living, however, additional therapeutic interventions may be necessary. These may include medication, surgical procedures, and hormonal therapy.

Several lines of evidence support the idea that energy balance is primarily regulated by the neuroendocrine system (NES), a complex system that integrates neuronal and hormonal signals [5,6]. Growth hormone (GH), also referred to as the “master hormone”, exerts regulatory control over metabolic homeostasis and exerts multifaceted effects on numerous physiological processes.

GH has been demonstrated to increase metabolic rate and improve insulin sensitivity, thereby regulating blood glucose levels and reducing the risk of developing obesity-related health complications, including T2DM [7,8,9,10]. While GH stimulates hepatic gluconeogenesis and glucose production [11,12,13], it induces insulin resistance and decreases glucose uptake in skeletal muscles [14,15,16]. Additionally, GH promotes lipolysis, which reduces fat mass and improves insulin sensitivity. Interestingly, and in apparent opposition, a mouse model in which GHR was ablated in white adipocytes exhibited increased systemic insulin sensitivity despite favoring body fat accumulation [17]. While the precise mechanisms by which GHR ablation leads to increased insulin sensitivity in white adipocytes are still being investigated, it is hypothesized that changes in adipokine secretion or alterations in lipid metabolism may be involved.

GH production is primarily controlled by the opposing actions of hypothalamic growth hormone-releasing hormone (GHRH) and somatostatin (SST), as well as the feedback mechanism that is mediated by insulin-like growth factor 1 (IGF-1) [5,18]. GH is secreted by the pituitary gland and binds to specific receptors on target cells, initiating a cascade of intracellular events. This signaling pathway activates various signaling molecules, such as Janus kinase (JAK), and signal transducers and activators of transcription (STAT), leading to the activation of downstream pathways that are involved in cell proliferation, differentiation, and survival. GH signaling also influences the production and release of IGF-1, which further mediates its effects on target tissues (see Figure 1).

IGF-1 has a high degree of structural similarity to insulin and binds strongly to the IGF-1 receptor (IGF-1R), activating both the mitogen-activated protein (MAP) kinase and phosphoinositide 3-kinases (PI3K) signaling pathways in the target tissue [19,20]. The majority of circulating IGF-1 originates from hepatic cells, and its production and release are controlled by GH [21].

In addition to its potential to decrease fat mass, GH treatment has been explored for its therapeutic potential in various diseases. For instance, GH deficiency (GHD) has been associated with increased cardiovascular mortality and decreased quality of life [22]. GH replacement therapy has been shown to improve cardiac function, body composition and, to some extent, the overall well-being of GHD patients [1,10]. GH treatment has also been investigated for its potential in managing metabolic disorders such as T2DM and metabolic syndrome [1,10]. Although the effects of GH on glucose metabolism are still controversial, some studies have shown improvements in insulin sensitivity and glucose homeostasis as a result of GH treatment in T2DM patients [23]. Therefore, GH therapy may have potential therapeutic implications in managing various diseases beyond obesity.

Given the rising prevalence of obesity and its related comorbidities, there is an urgent need for effective strategies for managing this condition. Modulating the GH and IGF-1 axis may offer a promising solution, as these hormones have been shown to regulate metabolic processes and body composition. However, further research is required to fully understand the intricate relationship between GH, IGF-1, and metabolism. Furthermore, the safety of GH therapy requires careful consideration due to its ability to induce anabolic effects by stimulating protein synthesis and cell proliferation [11,24,25].

It is important to note that animal models may not necessarily be directly applicable to humans, and more clinical trials are needed to fully evaluate the potential benefits and risks of GH therapy in managing obesity. Therefore, caution should be exercised in extrapolating the results of animal studies to humans, and further research is needed to determine the safety and efficacy of GH therapy in human populations.

This review aims to evaluate the therapeutic potential of targeting the GH and IGF-1 axis in the management of obesity by examining their interaction with metabolism.

## 2. Method

In this review, we conducted a comprehensive literature search using electronic databases, including PubMed, Embase, and the Cochrane Library. We used the following search terms: “GH”, “IGF-1”, “obesity”, “body weight”, “body composition”, “energy expenditure/balance”, and “metabolic disorders”. We limited our search to articles that were published in English from the year 1993 to 2023. We included randomized controlled trials, observational studies, transgenic mouse models, and systematic reviews that investigated the effects of GH and/or IGF-1 on body weight, body composition, and metabolic disorders. We excluded studies that focused on populations other than individuals with obesity or those that investigated other interventions in addition to GH and IGF-1. In addition to the current literature, we also referenced previous studies that have significantly impacted the current understanding of the role of GH and IGF-1 in obesity. These references date back to the early studies on mouse models and their application to human subjects. Furthermore, we have incorporated preliminary observations that were generated from our own newly developed models that are still undergoing thorough investigation. These observations provide additional insights into the effects of GH and IGF-1 on body weight, body composition, and metabolic disorders, augmenting the existing literature. The inclusion of our model-generated data adds to the breadth of evidence considered in this review and contributes to the ongoing exploration of the topic.

## 3. Organ Involvement in the Regulation of GH and IGF-1 Axis in Obesity

### 3.1. The Hypothalamus and Pituitary Gland (Hypothalamic–Pituitary Axis)

The hypothalamus is a crucial regulatory organ that integrates the nervous and endocrine systems; it plays a vital role in mediating physiological processes such as reproduction, somatic growth, energy balance, and metabolic homeostasis [26,27]. Although relatively small, the hypothalamus comprises less than 2% of the total brain and is located in the lower part of the diencephalon. It receives diverse signals from other brain regions and subsequently initiates behavioral responses to various environmental stimuli [4]. The hypothalamus plays a crucial role in regulating the endocrine system by releasing hormones into circulation, which travel to other endocrine glands to control their hormone production. The hypothalamus communicates with the pituitary gland through two distinct pathways. In the first pathway, neurosecretory cells of the hypothalamus synthesize hormones such as oxytocin (OT) and antidiuretic hormone (ADH), which are directly transported to the posterior pituitary gland via extended neuron fibers and axons. In the second pathway, the hypothalamus secretes hormones that are produced and stored in neuroendocrine cells in the hypothalamus, which are then transported to the anterior lobe of the pituitary gland via the hypophyseal portal system [4,28].

The hypothalamus receives and integrates signals from hormones, nutrients, and other factors to regulate appetite and metabolism through the actions of specific neuronal populations. The hypothalamus is a crucial regulatory center in the control of energy balance, and its medial portion, which is composed of the ventromedial nucleus (VMN), arcuate nucleus (ARC), and paraventricular nucleus (PVN), plays a key role in this process [29]. The ARC contains two distinct populations of neurons, one that produces orexigenic peptides such as agouti-related protein (AgRP) and neuropeptide Y (NPY) and another that secretes anorexigenic peptides, including proopiomelanocortin (POMC) and cocaine- and amphetamine-related transcript (CART) [30]. POMC neurons secrete several peptides, including the anorectic peptide alpha-melanocyte-stimulating hormone (α-MSH), which acts via melanocortin 4 receptor (MC4R) to reduce appetite and food intake [31,32]. The PVN also expresses melanocortin receptors 3 and 4 and NPY receptors, and it secretes neuropeptides such as corticotropin-releasing hormone (CRH), which exerts an anorexigenic action [4,33]. Furthermore, afferent signals, including leptin from adipose tissue, insulin from the pancreas, ghrelin, and peptide YY from the gastrointestinal tract, modulate the anorexigenic centers of the hypothalamus [4]. Output signals from the PVN and lateral hypothalamus (LH) activate the sympathetic nervous system or the vagus nerve via the autonomic nervous system. These signals also include the release of GHRH, SST, and thyrotropin-releasing hormone (TRH), which regulate the metabolism of adipose tissue and overall metabolic rate by controlling the secretion of pituitary hormones such as adrenocorticotropic hormone, GH, and thyroid-stimulating hormone (TSH) [4].

The hypothalamus is a crucial organ for regulating body weight as it regulates the balance between food intake, energy expenditure, and body fat storage. This is evident, because the majority of genetic syndromes of severe obesity are caused by mutations in genes that are expressed in the hypothalamus [34]. Hypothalamic obesity (HO) develops due to dysfunction in the hypothalamic regulatory centers of body weight and energy expenditure. HO may be due to structural damage to the hypothalamus, radiation therapy, genetic disorders such as Prader–Willi syndrome, and mutations in specific genes, such as LEP, LEPR, POMC, MC4R, and CART [35].

The hypothalamus also regulates energy expenditure by controlling the sympathetic nervous system and the activity of hormones such as thyroid hormone and insulin, thereby playing a crucial role in maintaining energy balance and body weight homeostasis [36].

The pituitary gland, also known as the hypophysis, is a small endocrine gland located at the base of the brain; it plays a crucial role in regulating physiological processes by secreting hormones that travel to other endocrine glands and organs in the body. The pituitary gland is divided into two main regions, the anterior lobe and the posterior lobe. The anterior pituitary develops from the oral ectoderm during embryonic development [37]. This gland is encased by a network of blood capillaries originating from the hypothalamus, forming the hypophyseal portal system. This system is responsible for conveying neuroendocrine signals from the hypothalamus to the anterior pituitary and subsequently from the anterior pituitary to the circulatory system. The anterior lobe, also referred to as the adenohypophysis, produces six hormones: GH, prolactin (PRL), thyroid-stimulating hormone (TSH), melanin-stimulating hormone (MSH), follicle-stimulating hormone (FSH), and luteinizing hormone (LH). The posterior lobe, or neurohypophysis, is derived from neuro-epithelial cells and is thus structurally and anatomically separate from the anterior lobe of the pituitary gland [38]. The posterior lobe contains neuro-glial cells and nerve fibers that extend from the hypothalamus, and it is considered an extension of the brain [27]. The posterior lobe secretes two hormones, OT and ADH, which are produced by neurosecretory cells in the hypothalamus and transported via axons to be stored in the posterior lobe. These hormones are then secreted into the circulatory system under the control of the hypothalamus [38]. The pituitary gland plays a role in the regulation of energy expenditure and body weight in obesity [39]. The pituitary gland’s anterior lobe produces several hormones that are involved in energy metabolism, including GH and TSH. GH is known to increase energy expenditure by promoting the breakdown of fat and by stimulating the production of IGF-1, which also promotes lipolysis under special circumstances [18,40,41]. TSH plays a crucial role in regulating the metabolic processes that are essential for normal growth and development, as well as metabolic homeostasis in adults. Thyroid hormone status is closely linked to body weight and energy expenditure [42]. Hyperthyroidism, characterized by an excess of thyroid hormone, leads to a hypermetabolic state that is characterized by increased resting energy expenditure, weight loss, decreased cholesterol levels, increased lipolysis, and increased gluconeogenesis [43].

OT and ADH, which are released from the posterior lobe of the pituitary gland, are also involved in the regulation of appetite and energy balance. In obesity, the balance between the secretion of these hormones can be disrupted, leading to an increase in appetite and a decrease in energy expenditure, contributing to weight gain [39].

### 3.2. The Liver and Adipose Tissue

The liver and adipose tissue serve as key regulators of whole-body energy homeostasis by coordinating glucose, lipid, and energy metabolism [44]. Insight into the roles of the liver and adipose tissue in metabolic homeostasis may provide a potential intervention strategy to alleviate the detrimental effects of obesity on human health. The liver is a relatively large organ that represents approximately 2–3% of the total body weight in humans, making it the second largest organ after the skin [45]. The liver is protected by the ribcage and is encased by peritoneal reflections [45]. The liver receives its blood supply from two distinct sources. The majority, approximately 80%, is supplied by the portal vein, which carries nutrient-rich blood from the spleen and intestines; the remaining 20% of the blood supply is oxygenated blood that is delivered by the hepatic artery [45,46]. Anatomically, the liver is divided into several functional regions, including the right and left lobes, which are separated by the falciform ligament, and the caudate and quadrate lobes, which are separated by the ligamentum venosum. The liver is also divided into functional units called lobules, which are hexagonal and consist of hepatic cells or hepatocytes, the primary functional cells of the liver [46]. Histologically, the liver is composed of several different cell types, including hepatocytes (HCs), hepatic stellate cells (HSCs), Kupffer cells (KCs), and liver sinusoidal endothelial cells (LSECs). Hepatocytes, which make up about 60–70% of the liver, are the primary functional cells of the liver and are responsible for most of the metabolic activity of the liver [47]. Sinusoidal endothelial cells line the blood sinusoids and regulate the flow of blood and nutrients through the liver [38,39,40].

Adipose tissue, commonly known as fat tissue, is a type of connective tissue with the primary function of storing energy in the form of lipids; it is composed of adipocytes, specialized cells that are capable of expanding and contracting to store and release fatty acids as necessary [48]. Adipose tissue is able to adapt and change in response to internal and external signals and can expand up to 15 times its original size [49]. Adipose tissue is a unique organ in that it possesses the ability for unlimited growth potential at any stage of life. The adipose tissue mass size is determined by the quantity of adipocytes present and the size of each cell. Adipose tissue expansion can occur through two distinct mechanisms: hyperplasia, or an increase in the number of adipocytes; and hypertrophy, an increase in the size of individual adipocytes. While hypertrophy, which is primarily due to lipid accumulation within the cell, is reversible, hyperplasia is a permanent change that persists throughout the individual’s lifetime [50]. Adipocytes are primarily composed of triglycerides, which are molecular structures that are formed by bonding three fatty acid molecules to a glycerol molecule. These triglycerides are stored in the cell as droplets, occupying up to 90% of the cell’s volume. Adipose tissue is distributed throughout the body, including in subcutaneous tissue, the retroperitoneal space, and in close proximity to organs such as the heart. Additionally, it envelops blood vessels and nerves and performs important functions such as insulation and cushioning [51]. The distribution of adipose tissue can vary among individuals and is influenced by various factors such as diet, genetics, and physical activity. It is also associated with various metabolic disorders such as obesity and T2DM. It has been linked to an increased risk of several chronic diseases, including cardiovascular disease and certain types of cancer [51,52]. In mammals, there are three main classes of adipose tissue: white adipose tissue (WAT), brown adipose tissue (BAT), and beige adipose tissue [53]. WAT is the most abundant and well-known type, composed of white adipocytes that store excess energy as triglycerides and provide insulation to regulate body temperature [54]. Conversely, BAT is composed of brown adipocytes that generate heat through thermogenesis, and they play a crucial role in regulating body temperature [55]. Beige adipose tissue, also known as “brown-in-white”, is a newly discovered type of adipose tissue that has the properties of both WAT and BAT. Beige adipocytes, interspersed among white adipocytes, undergo a phenotypic shift from a white-like state to a brown-like state in response to stimuli such as cold exposure or specific hormonal signals [56].

Extensive research has revealed that adipose tissue not only regulates glucose and lipid metabolism but also functions as an endocrine organ by releasing a variety of hormones and signaling molecules that regulate a wide range of physiological processes. These include energy expenditure, appetite control, insulin sensitivity, inflammation, and tissue repair [57]. Both WAT and BAT play important roles in the secretion of hormones and signaling molecules in the form of peptides, lipids, and microRNAs. These include leptin and adiponectin, which have potent roles in mediating metabolic processes [57].

Adipose tissue acts as a communication hub between various organs and the central nervous system, regulating energy supply and demand through hunger and satiety signals. Adipose tissue responds to insulin by converting glucose to lipids and subsequently storing the lipids as a reserve for future energy requirements. Additional signals that functionally modulate adipose tissue include sex hormones, which partially determine fat distribution in the body and mediate inflammatory responses [58]. Dysregulation of these functions leads to the development of metabolic diseases [59]. The accumulation of excess WAT in the abdominal region also plays a role in the development of obesity.

In obesity, adipose tissue becomes enlarged and dysfunctional, leading to a chronic low-grade inflammation state and an increased secretion of pro-inflammatory cytokines such as TNF-alpha and IL-6, which also contribute to the development of metabolic disorders such as T2DM, hypertension, and cardiovascular disease [48,60,61].

In summary, the liver and adipose tissue are two key organs that play a crucial role in the development and progression of obesity. The liver regulates metabolic processes, including glucose and lipid metabolism. Excess nutrient intake results in the development of hepatic steatosis (fatty liver), which is often seen in obesity. The resulting insulin resistance contributes to the development of metabolic diseases such as T2DM and hypertriglyceridemia [62]. Additionally, fat accumulation in the liver may lead to inflammation and oxidative stress, which contribute to the development of liver fibrosis [63].

## 4. The Advancements in the Understanding GH Secretion: Mechanisms, Modulators, and Consequences

The development of a reliable immunological assay for measuring circulating hormones represented a crucial advancement in endocrinology. This led to the identification of a previously unknown pulsatile pattern of GH secretion, providing a deeper understanding of GH’s physiological role and offering novel therapeutic avenues for GH deficiency-associated conditions [64,65]. The interaction between GH and IGF-1 is intricate and bidirectional, with GH promoting the synthesis and secretion of IGF-1 and IGF-1, subsequently inhibiting the secretion of GH from the pituitary gland through a negative feedback mechanism [5,66].

The mechanism of GH production is a complex process that is regulated by a variety of hormones and neurotransmitters. In normal individuals, GH is produced and released by the pituitary gland in a pulsatile manner. The primary triggers for GH release are the hypothalamic hormones GHRH and ghrelin. Ghrelin, a hormone produced by the stomach, promotes hunger and increases GH release, while GHRH stimulates the pituitary gland to release GH [67,68,69]. In addition, other hormones such as IGF-1 and cortisol modulate GH secretion. The release of GH is also suppressed by the hypothalamic hormone SST, which acts as an inhibitor for the release of GH [70]. Obesity has been associated with alterations in the pattern of GH production; however, the underlying mechanisms are not fully understood. Studies have shown that individuals with obesity have lower GH levels and a less pulsatile pattern of secretion than those with normal body weight [65]. A previous study has found that obesity impairs sensitivity to the effects of arginine infusion on human GH levels, with peak levels being significantly lower in obese subjects compared with normal subjects. However, changes in insulin levels were similar between the two groups [71,72]. A recent study was designed to investigate the impact of body composition on GH production in non-obese adults. The results showed that abdominal fat was the major predictor of GH production, with higher levels being associated with lower GH production. This study also found that physical fitness was positively associated with GH production. This study concluded that abdominal fat plays a significant role in determining GH secretion in healthy non-obese adults; however, the exact mechanisms behind this relationship are not yet understood [73]. In an attempt to understand the underlying mechanisms regulating GH production in obesity, scientists have proposed several hypotheses. One possible explanation is that excess adipose tissue in obesity may lead to increased levels of circulating insulin and free fatty acids (FFAs), which can inhibit GH secretion by suppressing the principal regulator, the GHRH hormone. A previous study aimed to investigate the effect of FFA on GH secretion in response to GHRH in a sample of six young, healthy men. The study found that the peak GH response following GHRH administration was significantly reduced in the group that received the lipid-heparin infusion compared with the peak GH response in the control group. These results suggest that elevations in plasma FFA levels can inhibit GH secretion in response to GHRH. Due to the small sample size, further research is necessary to fully understand the underlying mechanisms and generalizability of the findings [74]. Furthermore, pharmacological interventions that reduce FFA levels have increased GH secretion. Conversely, elevations in FFA levels have been found to inhibit GH secretion, both at baseline and in response to various stimuli, such as reductions in FFA levels, hypoglycemia, physical exercise, or administration of GHRH or GHRP-6 (a synthetic hexapeptide that specifically triggers the release of GH by pituitary somatotrophs) [75]. Additionally, excess adipose tissue may also lead to increased levels of circulating insulin and FFAs, inhibiting GH secretion [76].

Aging is associated with a decline in the production of GH and IGF-1, which can result in a host of negative consequences. Specifically, the decline in GH and IGF-1 can lead to decreased muscle mass, increased body fat, and reduced bone density [77]. These changes can negatively impact an individual’s quality of life, including causing an increased risk of falls and fractures. Additionally, decreased GH and IGF-1 levels have been linked to an increased risk of developing chronic conditions such as diabetes and heart disease [78]. Therefore, understanding the role of GH and IGF-1 in aging and the associated physiological changes are important for developing interventions to promote healthy aging.

## 5. The Negative and Positive Feedback Regulating GH Production

The GH–IGF axis is a complex system that plays a crucial role in regulating growth and metabolism [1,17,66]. The axis involves the interplay of several steroid hormones and proteins, which not only include GH and IGF-1 but also upstream signaling neuropeptides such as SST and GHRH, as well as the downstream signaling targets known as IGF-binding proteins (IGFBPs) [65,79,80,81]. This section will focus on the mechanisms that regulate the negative and positive feedback effects in the GH–IGF axis.

The GH–IGF axis operates through a positive and negative feedback loop, whereby GH, which is produced by somatotroph cells in the pituitary gland, stimulates the liver to produce circulating IGF-1 (see Figure 1). This process is mediated by a pathway that leads to the activation of the transcription factor, signal transducer and activator of transcription 5 (STAT5) (see Figure 1), which in turn stimulates the expression of IGF-1 [82]. The increased level of IGF-1 promotes cell growth and division by binding to its receptors on the cell surface and activating the PI3K/Akt signaling pathway [5,83,84].

One of the earliest studies that aimed to investigate the molecular mechanism by which IGF-1 suppresses GH gene expression at the pituitary level was conducted at Nagoya University in Japan. In that study, the researchers established a rat somatotroph tumor cell line, MtT/S, and transfected plasmids containing the GH 5′ promoter fused to the luciferase reporter gene. They found that IGF-1 suppressed GH promoter activity in a time and dose-dependent manner to inhibit GH secretion. Further experiments using deletion mutants of the GH promoter revealed that negative regulation was maintained on the shortest construct. This suggests that the IGF-1-related factor acts at the region nearest the minimal promoter. The study also found that the negative effect was eliminated by a PI3K inhibitor, indicating that the PI3K-mediated signaling pathway plays a major role in the negative regulation of IGF-1 [85].

A study was conducted to investigate how IGF-1 negatively regulates GH gene expression at the promoter level using a somatotroph cell line. The results revealed that IGF-1 inhibits GH mRNA levels by disrupting the binding of POU1F1 to the GH promoter through the inhibition of cyclic AMP (cAMP) response element binding protein (CBP). To confirm CBP’s role as a target of IGF-1R signaling, researchers used a mutant CBP construct and knock-in mouse model, which showed elevated serum GH levels, a greater response to GHRH stimulation, lower weight gain, and decreased body fat. The findings suggest that IGF-1R signaling disrupts the POU1F1/CBP complex to inhibit gene expression. Furthermore, chromatin immunoprecipitation assays demonstrated the inhibition of CBP binding to the GH promoter after IGF-1 treatment. Using a mutant CBP construct that lacked a critical phosphorylation site led to the loss of IGF-1 inhibition, supporting the hypothesis. The study confirmed the inhibitory effects of IGF-1 on GH expression at the promoter level and provided evidence of CBP’s role as a target of IGF-1R signaling [86].

To further understand the role of IGF-1 in negative feedback, our group developed a mouse model, referred to as the SIGFRKO mouse, in which the IGF-1R was ablated from the somatotroph and where they used the GH promoter as a driver for Cre recombinase. The SIGFRKO mice had increased GH gene expression and secretion and increased serum IGF-1. Additionally, the SIGFRKO mice had decreased GHRH and increased STT mRNA expression levels. These results support the idea that IGF-1 negatively regulates somatotroph synthesis and GH secretion, suggesting that hypothalamic feedback limits the extent of GH release [18].

A recent study investigating the role of IGF-1 in regulating the negative feedback of GH secretion developed a different mouse model in which the IGF-1R was specifically ablated in GHRH-expressing cells [84]. As expected, this mouse model presented an interesting phenotype that was characterized by an increase in GH levels, pulse amplitude, and frequency, as well as increased GHRH mRNA levels, GH mRNA expression, and serum IGF-1 levels, which were mediated by the pronounced elevation in the circulating level of GH. Despite the established role of GH in promoting lipolysis, this mouse model study did not demonstrate any effect on total fat mass despite the significant elevation in circulating GH levels [18]. Recently, our laboratory has developed a new mouse model, referred to as the S-GIGFRKO mouse, which is characterized by the ablation of the IGF-1R in both the somatotrophs and GHRH-expressing neurons. The results of our study revealed that the S-GIGFRKO mouse line displayed a modest increase in circulating GH levels and circulating IGF-1 levels. Furthermore, the ablation of IGF-1R in this mouse model was associated with increased lipolysis activity and a decrease in total body fat mass. These findings demonstrate the crucial role of IGF-1 in regulating GH production through negative feedback mechanisms [5]. Further details and analyses of these models will be provided in the next section of our study.

In conclusion, the GH–IGF axis is a complex system that is tightly regulated to maintain the balance between growth and metabolic processes. SST, GHRH, and IGFBPs play important roles in regulating the positive and negative feedback in the axis, ensuring a homeostatic response to changing physiological conditions. These mechanisms are mediated by different signaling pathways and receptors that are activated by interacting hormones and proteins. Further research is required to fully understand the intricacies of this system and its potential therapeutic applications.

## 6. Overview of the Current Understanding of the Relationship between the GH–IGF-1 Axis and Obesity

GH and IGF-1 are endocrine signaling molecules that are critical for regulating processes such as growth, maturation, and metabolic homeostasis. It has been suggested that deviations from normal GH and IGF-1 levels are linked with the development of obesity; however, the precise mechanisms underlying this relationship have yet to be fully elucidated [65,80,87].

One potential link between the GH–IGF-1 axis and obesity is the molecules’ effects on insulin sensitivity. IGF-1 has a nearly 50% amino acid sequence similarity with insulin and elicits an almost identical hypoglycemic response [88,89]. The capability of IGF-1 to bind to insulin receptors suggests its involvement in mediating insulin activity. Several experimental models have been used to demonstrate the effect of IGF-1 on insulin sensitivity and resistance.

Researchers have created a mouse model known as the liver IGF-1-deficient (LID) mouse to investigate the metabolic effects of IGF-1 deficiency. The LID mice showed significantly reduced levels of IGF-1 and elevated levels of GH in their circulation, which is associated with higher insulin levels and abnormal glucose clearance after insulin injection. However, their fasting blood glucose levels and levels after a glucose tolerance test appeared to be normal. This suggests that the LID mice are insulin resistant but can maintain normal blood glucose levels due to the high insulin levels in their circulation. Treatment with recombinant human IGF-1 or a GH-releasing hormone antagonist, which reduces GH levels, improved insulin sensitivity in the LID mice. These findings indicate that circulating IGF-1 plays a role in insulin action in peripheral tissues (Figure 2) [90].

Evidence suggests that GH and IGF-1 could contribute to the onset of obesity by impacting inflammation and oxidative stress. A previous study aimed to investigate the potential role of GH and IGF-1 in the development of obesity and focused on their role in mediating oxidative stress and inflammation. The study determined the effects of long-term HFD-induced obesity on vascular function and metabolic alterations in a Lewis dwarf rat model of GH/IGF-1 deficiency. The results show that GH/IGF-1 deficiency exacerbates vascular dysfunction, inflammation, and oxidative stress when challenged with an HFD. However, GH/IGF-1 deficiency did not affect weight gain or changes in body composition in response to the HFD challenge. Instead, the low insulin levels observed in the GH/IGF-1 deficient rats may be due to compromised β-cell numbers or function and impaired β-cell compensation in response to metabolic challenges. GH/IGF-1 deficiency was associated with increased adiponectin levels but normal serum levels of leptin. In the control animals, the HFD stimulated an inflammatory response to increase circulating levels of multiple inflammatory cytokines, including IL-6 and TNF-α; however, the exact mechanism is not well understood [91].

Another possible mechanism linking the GH–IGF-1 axis and obesity is through the effects on muscle mass and function and the development of adipose tissue. GH and IGF-1 are known to promote the growth and development of skeletal muscle, and low levels of these hormones have been associated with a reduction in muscle mass and function [92]. This may lead to an impaired ability to burn calories, potentially contributing to weight gain. In addition, previous studies using several transgenic mouse models have demonstrated the crucial role of the GH–IGF-1 axis in regulating adipocyte proliferation and differentiation [66,93]. Adipocyte differentiation is a complex process involving the activation of multiple transcription factors, signaling pathways, and epigenetic modifications [94]. The process starts with the commitment of mesenchymal stem cells to the adipocyte lineage, followed by the formation of preadipocytes. Preadipocytes then undergo a series of morphological and biochemical changes to become mature adipocytes, which are characterized by the accumulation of triglycerides and the expression of adipocyte-specific genes, such as peroxisome proliferator-activated receptor gamma (PPARγ) and adiponectin [95]. Finally, GH and IGF-1 may have effects on brain regions controlling the appetite [81,96]. Dysregulation of GH and IGF-1 may therefore disrupt normal appetite control and contribute to weight gain.

The relationship between GH, IGF-1, and obesity is complex and not fully understood (summary in Table 1). Further research is needed to fully understand the mechanisms by which these hormones may be linked to the development of obesity and to identify potential strategies for preventing and treating obesity.

## 7. Mouse Models Were Used to Study the Role of GH and IGF-1 in Obesity

The levels of circulating GH and IGF-1 exhibit an age-dependent pattern, with a gradual increase following birth, reaching a plateau during puberty and a subsequent decline of approximately 14% per decade of life [103,104]. The GH–IGF-1 axis plays a crucial role in the regulation of energy balance in the body, and disruptions in this axis have been linked to the development of obesity. As mentioned earlier, obesity is a complex metabolic disorder that results from an imbalance between energy intake and energy expenditure [105,106]. To gain a better understanding of the role of the GH–IGF-1 axis in obesity, researchers have utilized various mouse models to investigate the effects of alterations in these factors in the tissues and organs that are involved in energy metabolism, such as the liver, adipose tissue, and muscles [66]. These models have been valuable tools for studying the mechanisms by which the GH–IGF-1 axis regulates energy balance and for identifying potential therapeutic targets for obesity.

### 7.1. The Snell Dwarf Mouse

The Snell dwarf mouse is a naturally occurring mouse model first identified at the Bussey Institution at Harvard University in the 1930s [107]. This model is characterized by a phenotype of dwarfism, which is caused by mutations in the Pit-1 gene, a transcription factor that regulates the production of GH, TSH, and PRL. The Snell dwarf mouse serves as a valuable tool for understanding the GH–IGF-1 axis, as studies have demonstrated that the lack of Pit-1 results in decreased GH and IGF-1 production, leading to the dwarfism phenotype. This mouse model has been extensively utilized to investigate the role of GH and IGF-1 in various physiological processes such as bone growth, metabolism, and longevity. Additionally, the Snell dwarf mouse has provided valuable insight into the mechanisms of Pit-1-dependent gene expression and its role in the regulation of GH and PRL production. The observations generated from this mouse model have helped to increase our understanding of the current modern science and have been widely used to shape the current research in the field.

### 7.2. The Ames Dwarf Mouse

The Ames dwarf mouse is a naturally occurring mouse model that was first identified in the 1960s. This model is characterized by a phenotype of dwarfism, which is caused by mutations in the Pit-1 gene. This mutation leads to a severe growth defect that results in proportionate dwarfism, with adult mice that are half the size of their control littermates. Additionally, Ames dwarf mice are sterile and hypothyroid. At the cellular level, Ames dwarf mice have an almost complete absence of pituitary somatotrophs, lactotrophs (PRL-producing cells), and thyrotropes (TSH-producing cells). This lack of these three cell types causes a lack of GH, PRL, and TSH in the mice. The PROP1 gene encodes a transcription factor, which contains a paired-like homeodomain. Ames dwarf mice have a serine-to-proline amino acid substitution in the DNA-binding domain of PROP1. This substitution results in the mutant PROP1 being unable to bind to DNA effectively; thus, the three pituitary cell types fail to differentiate and proliferate during development [108]. This mouse model is useful for understanding the GH–IGF-1 axis, as it has been shown that the lack of GH results in decreased IGF-1 production, leading to the dwarfism phenotype. In addition, Ames mice have been used to study the role of GH and IGF-1 in bone growth, metabolism, and longevity, as well as the effect of GH on brain development, behavior, and cognitive function. The Ames dwarf mouse has been shown to have an elevation in fat mass, making it a useful model for studying the effects of GH on obesity [109].

### 7.3. The Metallothionein-I Human Growth Hormone Transgenic Mouse Model (MT1-hGH)

In 1983, a collaboration project between the University of Washington and University of Pennsylvania resulted in the creation of the first transgenic mouse model associated with an alteration in the GH–IGF-1 axis. The scientists utilized the promoter or regulatory region of the mouse gene for metallothionein-I (MT1) and fused it with the structural gene coding for human growth hormone (hGH). Microinjection of these fusion genes into fertilized eggs resulted in the generation of transgenic mice (MT1-hGH). These transgenic mice exhibited increased size compared with control mice, which was attributed to the elevation in serum levels of GH, which subsequently resulted in an elevation in IGF-1. Additionally, the transgenic mice showed alterations in pituitary function, specifically in the form of a dysfunction of the cells involved in the synthesis of GH. This model has been extensively utilized to study the role of the GH–IGF-1 axis in growth and development and has contributed to a deeper understanding of the mechanisms’ underlying growth and development as well as the complex interactions between these hormones and their receptors. This mouse model has been instrumental for advancing research in the field and for the development of new therapies for growth disorders. This model has helped to illuminate several concepts that are still widely used in modern research today [110].

### 7.4. The Adult-Onset, Isolated, Growth Hormone Deficiency (AOiGHD) Mouse Model

The AOiGHD mouse model is a genetically engineered mouse model that is characterized by a deficiency of GH in adult mice. This model was developed by crossbreeding rat GH promoter-driven Cre recombinase mice (Cre) with inducible diphtheria toxin receptor mice (iDTR). The resulting Cre^+/−^, iDTR^+/−^ offspring were then treated with diphtheria toxin to selectively destroy the somatotroph population of the anterior pituitary gland, leading to a reduction in circulating GH and IGF-1 levels. The main goal of the study was to investigate the hypothesis that the decline of GH levels observed with weight gain and normal aging may contribute to metabolic dysfunction. The study aimed to understand the effects of GH on fat accumulation, protein accretion, and insulin sensitivity under different feeding conditions. The results of the study showed that AOiGHD mice improved whole-body insulin sensitivity in both low-fat and high-fat-fed conditions and that these mice preferentially utilized carbohydrates for energy metabolism. However, in high-fat-fed AOiGHD mice, the fat mass increased, hepatic lipids decreased, and glucose clearance and insulin output were impaired. These findings suggest that low GH in the context of excess caloric intake could contribute to the development of diabetes [104].

### 7.5. The GH^−/−^ Mouse Model

GH^−/−^ mice are characterized by a deficiency of GH. The method that was used to create these mice involved the removal of the entire gene coding region of the mouse GH genomic sequence using the VelociGene KOMP definitive null allele design, which involves replacing the removed sequence with a ZEN-UB1 reporter/selection cassette. These mice have a genetic deletion of the GH gene, resulting in a lack of GH production in the pituitary gland. GH^−/−^ mice exhibit a phenotype of reduced body size, muscle mass, and bone density, as well as increased fat mass, similar to observations in human patients with GH deficiency. This mouse model is useful for studying the effects of GH deficiency in various physiological processes, as well as for developing therapies for GH deficiency in humans. Studies using GH^−/−^ mice have been used to investigate the effects of GH on the aging-related decline in muscle mass, bone density, and metabolism. Additionally, this model has been employed to study the impact of GH deficiency on the brain and its effects on cognitive function and behavior [111].

### 7.6. The GHR^−/−^ (Laron) Mouse Model

The Laron syndrome mouse model, created by a team of researchers led by Dr. Kopchick at the University of Ohio in 1997 [112], is considered one of the major milestones in enhancing our understanding of the role of GH in metabolic homeostasis and growth development. This model was created to mimic Laron syndrome in humans, which is due to mutations in the GH receptor (GHR) gene and is unique, as GHR deficiency has only been reported in humans and not in any other mammals. This model presents severe postnatal growth retardation, proportionate dwarfism, decreased IGF-1 levels, and elevated serum GH concentrations. This mouse model has been extensively utilized in research to gain insight into the mechanisms of GH in growth and aging. Furthermore, it has won the Methuselah Mouse Prize, an award for the longest-lived mouse model (i.e., its lifespan was shown to extend to almost 5 years of age), thus making the Laron mouse the most significant contribution to aging research [113]. It also serves as a valuable model for understanding the pathogenesis of obesity and the role of the GH–IGF-1 axis in the regulation of energy metabolism.

### 7.7. The Adipocyte-Specific GHR Knockout (AdGHRKO) Mouse Model 

Adipocyte-specific GHR KO (AdGHRKO) mice were recently created by the same group of scientists who created the Laron mouse to better understand the role of GH signaling directly in adipose tissue. Using the Cre/Lox strategy to specifically ablate GHR expression from adipose tissue, the authors showed that AdGHRKO mice have increased adiposity but appear healthy with enhanced insulin sensitivity. Additionally, the AdGHRKO mice had increased fat mass; reduced circulating levels of insulin, c-peptide, adiponectin, and resistin; and improved frailty scores, with increased grip strength at advanced ages in both sexes. The study found that disrupting the GHR gene in adipocytes improved insulin sensitivity at an advanced age and increased lifespan in male AdGHRKO mice [111]. Overall, the results indicate that removing GH’s action, even in a single tissue, can have observable health benefits, promoting long-term health, reducing frailty, and increasing longevity. By specifically ablating GHR expression from adipose tissue, this mouse model can be utilized by researchers to study the specific effects of GH on energy metabolism and the development of obesity. The results obtained from this model could aid in identifying potential therapeutic targets for the treatment of obesity and related metabolic disorders.

### 7.8. The Somatotroph IGF-1R Knockout Mouse Model (SIGFRKO)

This mouse model was developed in 2010 by a team of scientists at John Hopkins University using the Cre/lox system to specifically delete the IGF-1R from the somatotroph cells. The SIGFRKO mouse was used to study the role of IGF-1 in regulating the expression and release of GH. The SIGFRKO mouse showed increased GH expression and secretion, as well as increased serum IGF-1 levels. Additionally, there were compensatory changes in the expression of GHRH and SST, and the mice had normal linear growth in adulthood. Metabolic studies also revealed an elevation in the metabolic activity associated with an elevation in energy expenditure, reducing total fat mass due to increased lipolytic activity. These findings support the notion of negative regulation of GH expression and release by IGF-1 and suggest that hypothalamic feedback plays a role in limiting GH release. The SIGFRKO mouse also serves as a valuable tool for understanding the mechanisms of IGF-1 regulation in the hypothalamic-pituitary axis as well as the compensatory mechanisms that mediate growth and metabolic function in mammals (see Figure 3) [18,40].

### 7.9. The Somatotroph GHRH Neurons IGF-1R Knockout (S-GIGFRKO) Mouse Model 

Recently, our laboratory has generated a novel transgenic mouse model that is characterized by the selective deletion of the IGF-1R in GHRH neurons and somatotrophs. This model was designed to investigate the role of IGF-1R signaling in the regulation of GHRH-mediated GH production and growth. The S-GIGFRKO mice exhibited a modest increase in serum GH levels and GH gene mRNA expression, as well as a modest increase in serum IGF-1 levels (Figure 4) [5]. A gene expression analysis revealed that the deletion of IGF-1R resulted in an elevation of GHRH and SST in the hypothalamus, suggesting a compensatory mechanism. The S-GIGFRKO mice appeared to grow normally, but adult mice had a reduction in weight gain compared with control littermates. A body composition analysis showed a reduction in total fat mass but no changes in lean mass. A metabolic analysis revealed an elevation in the metabolic activity associated with increased energy expenditure. These findings provide new insights into the role of IGF-1R signaling in GH production and growth regulation and the potential use of this mouse model for further research on GH-related disorders. This unique mouse model presents a robust system not only for uncovering the functional significance of IGF-1 in somatotrophs and the hypothalamus but also for understanding the role of the IGF-1R–GHRH pathway in the regulation of body weight and energy balance [5,114]. Nevertheless, caution should be exercised in extrapolating the results of animal studies to humans, and further research is needed to determine the safety and efficacy of GH therapy in human populations (Table 2).

## 8. The Effect of Obesity on GH and IGF-1 Production

Obesity is associated with a marked blunting of GH secretion, which is both spontaneous and is evoked by provocative stimuli. This reduction in GH secretion is observed in response to traditional pharmacological stimuli acting in the hypothalamus, such as insulin-induced hypoglycemia, arginine, galanin, L-dopa, clonidine, and acute glucocorticoid administration, as well as in response to direct somatotroph stimulation by exogenous GHRH [115].

The impact of obesity on serum IGF-1 levels is a matter of controversy within the scientific community. While some studies have shown no alterations in IGF-1 levels in obesity, others have indicated a decrease in IGF-1 levels in the presence of obesity; others have also demonstrated an increase in IGF-1 levels in obese individuals [65,116]. These seemingly contradictory findings may be explained by the high levels of insulin that are present in obesity, which has been shown to increase IGF-1 production in the liver while also reducing the formation of IGF-binding protein 1. This increased availability of free and active IGF-1, sustained by high insulin levels, may explain the decrease in GH secretion through negative feedback mechanisms. Therefore, understanding the mechanisms underlying the altered regulation of GH secretion in obesity is an important area of research as it may have implications for the treatment of obesity and related metabolic disorders.

Clinical trials have investigated the potential use of GH and IGF-1 as interventions for obesity. For example, A meta-analysis of 24 studies involving almost 500 obese individuals found that GH treatment led to a decrease in fat mass of about 1 kg and an increase in lean body mass of about 2 kg over an average of 12 weeks. The outcome of this study suggests that treatment with recombinant human rhGH leads to a reduction in visceral fat and an increase in lean body mass in obese adults without causing weight loss. However, the treatment also leads to increases in fasting plasma glucose and insulin levels. The study used relatively high doses of rhGH, and further studies with longer durations and lower doses are needed to better understand the effects of rhGH therapy on obesity and its potential impact on cardiovascular health [117]. In all clinical studies that use rhGH as a therapeutic agent, caution is urged as its therapeutic safety is assessed. In summary, more research is needed to fully understand the effects of GH therapy and its potential side effects.

## 9. Conclusions and Further Study

This review of the literature highlights the potential therapeutic benefits of targeting GH and IGF-1 in the management of obesity. The complex interplay between these hormones and growth, as well as their roles in regulating energy balance and body composition, suggest that modulating GH and IGF-1 levels may be a promising strategy for combating obesity and its associated comorbidities. However, caution should be exercised as there are several limitations and challenges that are associated with their use. The safety and efficacy of GH and IGF-1 as a treatment for obesity are not clear, and long-term studies have not been conducted. Furthermore, GH and IGF-1 are both hormones that are naturally present in the body, and their use as therapeutic agents can disrupt the body’s homeostatic mechanisms, leading to untoward side effects. These side effects can include joint pain, carpal tunnel syndrome, and an increased risk of diabetes and cancer.

Moreover, it is important to note that animal models may not necessarily be directly applicable to humans, and more clinical trials are needed to fully evaluate the potential benefits and risks of GH and IGF-1 therapy in managing obesity. While the results obtained from animal studies provide valuable insights, caution should be exercised in extrapolating these findings to human populations. Human physiology and response to treatment can significantly differ from animal models, and therefore, it is essential to conduct well-designed clinical trials involving human subjects. These trials will not only help determine the safety and efficacy of GH and IGF-1 therapy in humans but also provide more accurate information on the potential benefits and risks associated with these interventions in the context of obesity management. Only through comprehensive research involving human populations can we confidently assess the feasibility and suitability of GH and IGF-1 therapies as effective strategies for combating obesity.

Therefore, while GH and IGF-1 show promise as therapeutic agents for obesity, more research is needed to fully understand their effects and potential side effects. Randomized, controlled trials are needed to confirm the efficacy of GH and IGF-1 targeted therapies in treating obesity and determine their long-term safety. Additionally, the high cost and complexity of administration present challenges for their practical application. It is important to approach the use of GH and IGF-1 in obesity management with caution, considering the potential risks and limitations that are associated with their use.

In conclusion, GH and IGF-1 may offer a promising avenue for the management of obesity, but caution should be exercised, and more research is needed to fully understand their effects, develop safe and effective treatment strategies, and validate their findings in human populations.

## Figures and Tables

**Figure 1 ijms-24-09556-f001:**
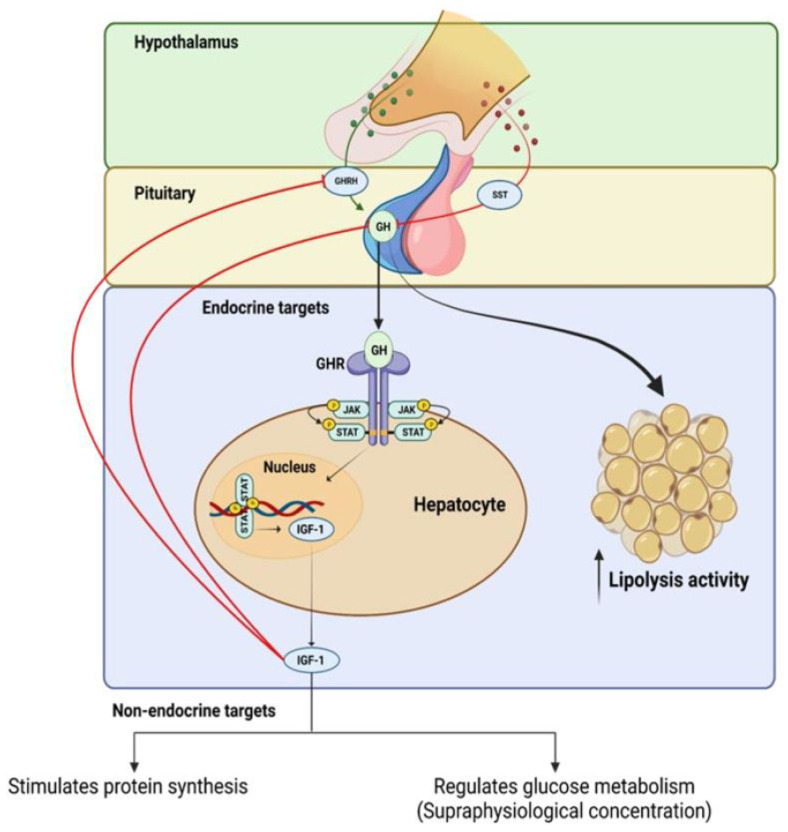
Positive and negative feedback regulation of growth hormone (GH) production and intracellular mechanisms of growth hormone receptor (GhR) signaling.

**Figure 2 ijms-24-09556-f002:**
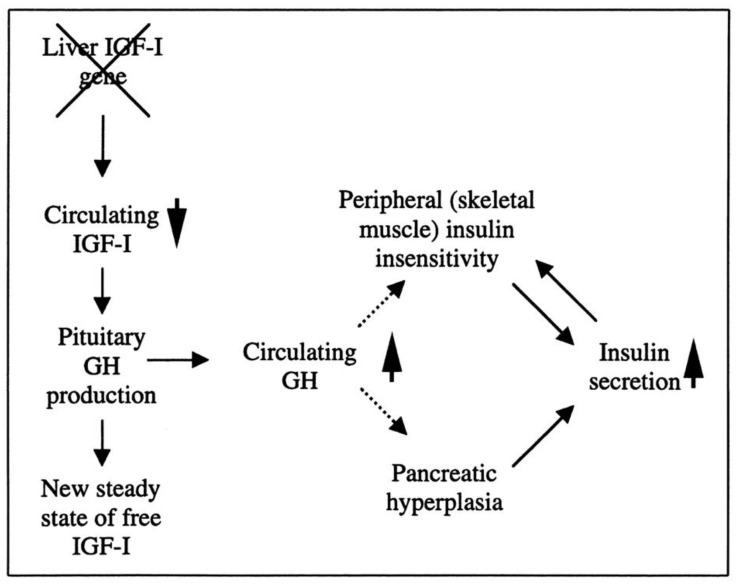
The role of liver-derived circulating IGF-I in muscle insulin sensitivity. Liver-specific igf-1 gene deletion results in reduced circulating total IGF-I and elevated GH levels, leading to insulin insensitivity in muscle and islet cell hyperplasia with hyperinsulinemia.

**Figure 3 ijms-24-09556-f003:**
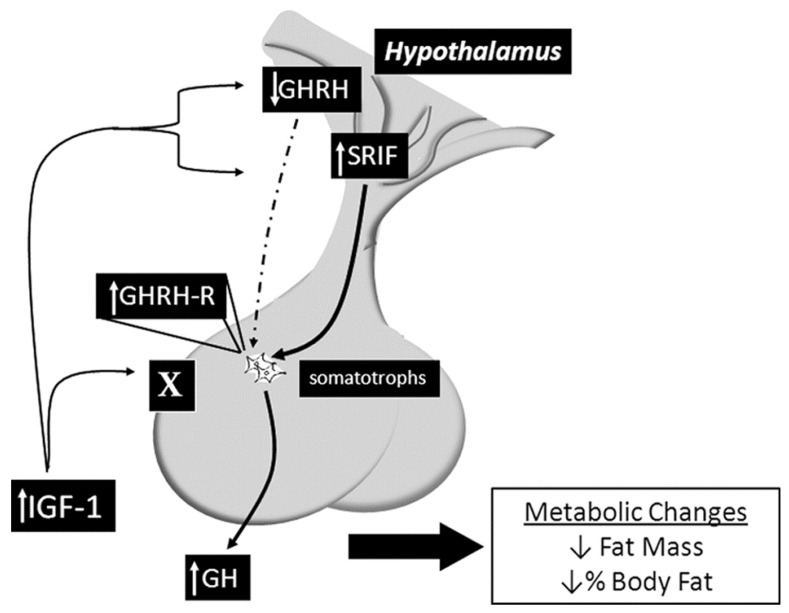
Summary of the main findings of the SIGFRKO mouse model, which lacks the IGF-1R gene in the somatotrophs, revealing altered signaling pathways and various physiological and pathological effects. Sourced from [18].

**Figure 4 ijms-24-09556-f004:**
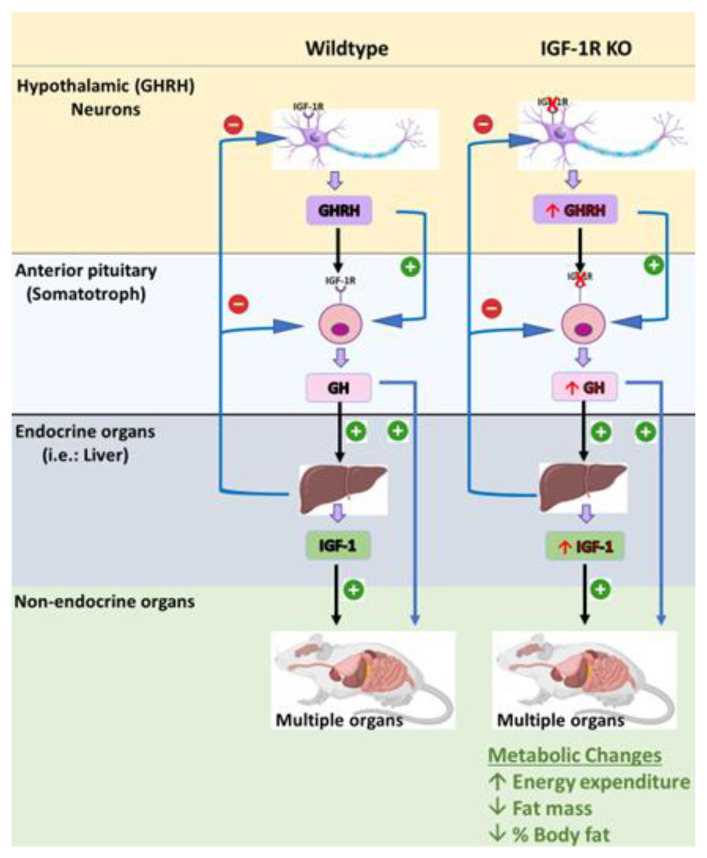
The S-GIGFRKO mouse model is characterized by selective ablation of the IGF-1R from the somatotrophs and GHRH neurons in the ARC. The model shows the effects of GH and IGF-1 on different physiological activities. Sourced from [5].

**Table 1 ijms-24-09556-t001:** Overview of the current understanding of the relationship between the GH–IGF-1 axis and obesity.

GH–IGF-1 Axis and Obesity	Summary of Current Understanding	References
GH deficiency and obesity	GH deficiency can lead to an increase in body fat mass, while GH replacement therapy can decrease it. However, the relationship between GH deficiency and obesity is complex and is influenced by various factors.	[97,98]
GH excess and obesity	GH excess is associated with decreased body fat mass but also with insulin resistance and glucose intolerance.	[99]
IGF-1 and obesity	IGF-1 levels are positively correlated with body fat mass, and low IGF-1 levels have been linked to obesity-related complications such as insulin resistance and T2DM. However, the relationship between IGF-1 and obesity is not fully understood and may be influenced by other factors such as age and gender.	[80,100]
GH–IGF-1 axis and adipocyte differentiation	The GH–IGF-1 axis plays a crucial role in regulating adipocyte proliferation and differentiation. GH promotes the differentiation of preadipocytes into mature adipocytes, while IGF-1 promotes the proliferation of preadipocytes. Dysregulation of this process may contribute to the development of obesity.	[94]
GH–IGF-1 axis and appetite regulation	GH and IGF-1 have been shown to affect appetite regulation through various mechanisms, such as stimulating the production of leptin and ghrelin. However, the exact role of the GH–IGF-1 axis in appetite regulation and its contribution to obesity is still unclear.	[101,102]

**Table 2 ijms-24-09556-t002:** Summary of mouse models used to study the GH–IGF-1 axis and obesity.

Mouse Model	Characteristics	Applications
Snell dwarf mouse	Naturally occurring mouse model with dwarfism caused by mutations in the Pit-1 gene, which regulates the production of GH, TSH, and PRL. Lack of Pit-1 results in decreased GH and IGF-1 production.	A valuable tool for understanding the GH–IGF-1 axis; extensively utilized to investigate the role of GH and IGF-1 in various physiological processes, such as bone growth, metabolism, and longevity.
Ames dwarf mouse	Naturally occurring mouse model with dwarfism caused by mutations in the Pit-1 gene, resulting in proportionate dwarfism with adult mice that are half the size of their control littermates. Almost complete absence of pituitary somatotrophs, lactotrophs, and thyrotropes.	Useful for understanding the GH–IGF-1 axis; studied the role of GH and IGF-1 in bone growth, metabolism, and longevity, as well as the effect of GH on brain development, behavior, and cognitive function; shown to have an elevation in fat mass, making it a useful model for studying the effects of GH on obesity.
MT1-hGH transgenic mouse	The transgenic mouse model was generated by fusing the promoter or regulatory region of the mouse gene for metallothionein-I (MT1) with the structural gene coding for human growth hormone (hGH). Exhibits increased size compared with control mice due to elevated levels of GH and a subsequent elevation in IGF-1.	Extensively utilized to study the role of the GH–IGF-1 axis in growth, development, and metabolism; has been used to investigate the effects of alterations in GH and IGF-1 in various tissues and organs involved in energy metabolism such as the liver, adipose tissue, and muscle.
(AOiGHD) mouse model	The model was developed by crossbreeding rat GH promoter-driven Cre recombinase mice with inducible diphtheria toxin receptor mice (iDTR) to selectively destroy the somatotroph population of the anterior pituitary gland, leading to a reduction in circulating GH and IGF-1 levels.	This model has been used in various studies to investigate the role of GH in metabolic regulation and to understand the mechanisms underlying metabolic disorders such as obesity, insulin resistance, and diabetes
GH^−/−^ mouse model	A genetically engineered mouse model that lacks GH production in the pituitary gland due to a genetic deletion of the GH gene. These mice exhibit a phenotype of reduced body size, muscle mass, and bone density, as well as increased fat mass.	This model is useful for studying the effects of GH deficiency in various physiological processes and for developing therapies for GH deficiency in humans.
The GHR^−/−^ (Laron) mouse model	A mouse model with a targeted disruption of the gene encoding the GH receptor (GHR), resulting in a lack of functional GHR.	Useful in understanding the role of the GHR in various physiological processes such as growth, metabolism, and immune function; provides insights into the mechanisms of GHR signaling and its downstream effects on energy metabolism.
(AdGHRKO) mouse model	This model was specifically designed to ablate GHR expression from adipose tissue, which caused increased adiposity, enhanced insulin sensitivity, increased fat mass, reduced circulating levels of insulin, c-peptide, adiponectin, and resistin, improved frailty scores with increased grip strength at advanced ages in both sexes, and increased lifespan in male AdGHRKO mice.	Study the specific effects of GH on energy metabolism and the development of obesity; identify potential therapeutic targets for the treatment of obesity and related metabolic disorders.
Somatotroph IGF-1R knockout mouse model (SIGFRKO)	Uses the Cre/lox system to specifically delete the IGF-1R from the somatotroph cells. Increased GH expression and secretion, as well as increased serum IGF-1 levels as compensatory changes in the expression of GHRH and SST; normal linear growth in adulthood, elevation in metabolic activity associated with an elevation in energy expenditure reducing the total fat mass due to increased lipolytic activity	A valuable tool for understanding the mechanisms of IGF-1 regulation in the hypothalamic–pituitary axis and the compensatory mechanisms that mediate growth and metabolic function in mammals.
Somatotroph GHRH neurons IGF-1R knockout (S-GIGFRKO) mouse mode	Selective deletion of the IGF-1R in GHRH neurons and somatotrophs resulted in a modest increase in serum GH levels and GH gene mRNA expression, as well as a modest increase in serum IGF-1 levels; elevation of GHRH and SST in the hypothalamus normal growth, but adult mice had a reduction in weight gain compared with control littermates; reduction in total fat mass but no changes in lean mass. Elevation in metabolic activity associated with increased energy expenditure	Provides new insights into the role of IGF-1R signaling in GH production and growth regulation and the potential use of this mouse model for further research on GH-related disorders. Can be used to understand the role of the IGF-1R–GHRH pathway in the regulation of body weight and energy balance.

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
