# Peer review of "Exploring the Therapeutic Potential of Targeting GH and IGF-1 in the Management of Obesity: Insights from the Interplay between These Hormones and Metabolism"

_ijms, 2023, doi:10.3390/ijms24119556_

Round 1

Reviewer 1 Report

Exploring the Therapeutic Potential of Targeting GH and IGF-1 in the Management of Obesity: Insights from the Interplay be- tween These Hormones and Metabolism

The abstract is pretty well-written and points out the key facts and outlines what the manuscript is going to cover

The first paragraph does a good job of providing useful statists on obesity

The second paragraph outlines key aspects of obesity and its management

The role of GH as a master hormone then ensues

GH has been shown to increase metabolic rate and improve insulin sensitivity, which can help regulate blood glucose levels and reduce the risk of developing obesity-related health complications such as type 2 diabetes (T2D) and then this sentence: A mouse model characterized by a GHR ablation in white adipocytes in- creases systemic insulin sensitivity, despite favouring body fat accumulation [18]. Can the authors justify why GH and then the ablation of GHR both led to insulin sensitivity?

GH production is primarily controlled by the opposing actions of hypothalamic growth hormone-releasing hormone (GHRH) and somatostatin (SST), as well as the feedback mechanism mediated by the insulin-like growth factor (IGF-1) [6, 20]. IGF-1 has a high degree of structural similarity to insulin and binds strongly to the IGF-1 receptor (IGF-1R) activating both the mitogen-activated protein (MAP) kinase and phosphoinositide 3-kinases (PI3K) signalling pathways in the target tissue [21, 22]. The majority of circulating IGF-1 originates from hepatic cells, and its production and release are controlled by GH [23].  This paragraph, which is key to understanding this manuscript, could benefit from a figure showing this signalling pathway.

Then the justification for the GH treatment ensues. I think that between this paragraph and the previous one, a key aspect is missing and that is the indication of GH in disease so that management can be envisaged. Please add a few sentences on this as well. I realise 2 paragraphs before you do mention GHR ablation but make a point about human disease.

In the figure indicated above (to add about the GH/ IGF1 axis), perhaps the mouse model you have developed could be depicted as well to help better understanding.

I really think that evidence on page 4 (the LID) mouse could benefit from adding a figure.

the ablation of the IGF-1R specifically in somatotroph cells referred to as the somatotroph-specific IGF-1R knock-out (SIGFRKO). This model demonstrated that the inhibition of the negative feedback mediated by IGF-1 on GH production resulted in the elevation of both circulating GH and IGF-1 levels which subsequently led to increased energy expenditure and lipolytic activity 

The results of the study demonstrated that while both groups of rats on a high-fat diet had similar weight gain, the GH/IGF-1 deficient Lewis Dwarf rats had elevated blood glucose levels, reduced insulin levels, and impaired glucose tolerance. Analysis of serum cytokine expression and aortic tissue revealed that GH/IGF-1 deficiency exacerbated inflammation, endothelial dysfunction, and oxidative stress in the GH/IGF-1 deficient Lewis Dwarf rats.  This is an interesting link in that the loss of the GH/IGF-1 axis has led to heightened inflammation, do the authors know by what mechanism this happens?

In addition, previous studies using several transgenic mouse models, have demonstrated the crucial role of the GH-IGF-1 axis in the regulation of adipocyte proliferation and differentiation. The authors could briefly mention the process of adipocyte differentiation.

Chapter 2 could benefit from a table to outline the key findings about the relationship between the GH and IGF-1 axis with obesity.

Quite a comprehensive section about the hypothalamus on page 6 and the pituitary gland on page 7. I am normally in favour of explaining things at great length. Still, it is up to you if you think this level of detail about these two glands is necessary (at the physiology textbook level), especially in section 2 you have been recounting peer-reviewed evidence from primary studies. It’s your call.

The comment above also applies to the liver/ adipose tissue section (3.2.). Given that these terms were used previously in an earlier chapter, this level of definition and detail here might be a little out of place but ultimately this is your manuscript. Perhaps these could be moved up to chapter 1 and then the following chapters could then deal with primary studies. The logical flow of the topics is not great at present.

The topic then returns to GH and IGF1.

Figure 1 is useful but could also incorporate STAT5. Again, the IGF1 and GH axis has been referred to in an earlier section (I think it was the introduction), so maybe one solid section on this could be provided and then that could get referred to in any subsequent mention (to consolidate these two sections to prevent repetition).

The primary study that follows then outlines the mechanism of the negative feedback of IGF-1 on GH via binding to its promoter. Another study using the MtT/S somatotroph cell line investigated the role of CREB-binding protein (CBP), required for the activation and regulation of GH gene expression, as a target of IGF-1 somatotroph regulation [82].  CREB binding by CBP requires the phosphorylation of CBP. In this regard, I am unsure what “required for the activation and regulation of GH gene expression, as a target of IGF-1 somatotroph regulation” means. I am aware that IGF1R can translocate to the nucleus, so what is binding the promoter of GH to regulate it? Is it P-CBP or IGF1R or some specific TF and co-actiavtor? Kindly make this clearer. Perhaps you could depict this in a figure. 

Another comment about the logical flow of this manuscript. The mouse model of the authors gets referred to multiple times and it might be better to consolidate this information. I have also looked up your previous review ( https://doi.org/10.3390/cells10102664) and the mouse model figure you have provided there is really informative and clear, perhaps something similar could be provided here so the authors can better understand the outcome of IGF-1R deletion (increased GHRH, GH and IGF-1).

Despite the established role of GH in promoting lipolysis, this mouse model study did not demonstrate any effect on total fat mass despite the significant elevation in circulating GH levels [20].  The authors could explain why this is the case and whether a parallel mechanism might be kicking in.

Recently, our laboratory has developed a new mouse model, referred to as the S- GIGFRKO mouse, which is characterized by the ablation of the IGF-1R in both the somatotrophs and GHRH-expressing neurons. The results of our study revealed that the S- GIGFRKO mouse line displayed a modest increase in circulating GH levels and circulating IGF-1 levels.  Does this suggest that a negative regulatory input from knocking out IGF-1R in somatostatin cells? The authors could explain this more in the text. 

The topics in section 5 (various mouse models 5.1- 5-9) could be summarised in a table. This will allow easier cross-comparisons between the models for any future reader. This way the readers can more easily choose their model of choice for their specific experiment. 

Section 6. For this section, I would also suggest providing a figure about the contribution of these other hormones to the axis of GH/IGF-1.

I appreciate that the authors have added information about the clinical trials but the abstract has outlined that detail about the management of obesity from the viewpoint of the axis of GH/IGF-1 will be provided, is there any other type of information about management (other than clinical trials) that can be added?

Author Response

Dear Reviewer (1),

We would like to express our gratitude for your valuable time and effort in reviewing our manuscript titled "Exploring the Therapeutic Potential of Targeting GH and IGF-1 in the Management of Obesity: Insights from the Interplay between These Hormones and Metabolism." Your insightful and constructive comments have been instrumental in enhancing the overall quality of our manuscript.

  • GH has been shown to increase metabolic rate and improve insulin sensitivity, which can help regulate blood glucose levels and reduce the risk of developing obesity-related health complications such as type 2 diabetes (T2D) and then this sentence: A mouse model characterized by a GHR ablation in white adipocytes in- creases systemic insulin sensitivity, despite favouring body fat accumulation [18]. Can the authors justify why GH and then the ablation of GHR both led to insulin sensitivity?

Thank you for bringing this to our attention to help us improve the clarity of our review. Indeed, we can address your question and provide more clarification in the review. A new statement was added on page 5, lines 8-10.

  • GH production is primarily controlled by the opposing actions of hypothalamic growth hormone-releasing hormone (GHRH) and somatostatin (SST), as well as the feedback mechanism mediated by the insulin-like growth factor (IGF-1) [6, 20]. IGF-1 has a high degree of structural similarity to insulin and binds strongly to the IGF-1 receptor (IGF-1R) activating both the mitogen-activated protein (MAP) kinase and phosphoinositide 3-kinases (PI3K) signalling pathways in the target tissue [21, 22]. The majority of circulating IGF-1 originates from hepatic cells, and its production and release are controlled by GH [23]. This paragraph, which is key to understanding this manuscript, could benefit from a figure showing this signaling pathway.

We agree that adding a figure to illustrate the GH-IGF-1 axis will enhance reader understanding. A new figure was added to page 17.

  • Then the justification for the GH treatment ensues. I think that between this paragraph and the previous one, a key aspect is missing and that is the indication of GH in disease so that management can be envisaged. Please add a few sentences on this as well. I realise 2 paragraphs before you do mention GHR ablation but make a point about human disease.

Thank you for your valuable feedback. A new paragraph add to page 5, lines 18-27

  • In the figure indicated above (to add about the GH/ IGF1 axis), perhaps the mouse model you have developed could be depicted as well to help better understanding.

We agree that adding the mouse model we developed to the GH/IGF1 axis figure could be beneficial for better understanding. We included a figure on page 27.

  • I really think that evidence on page 4 (the LID) mouse could benefit from adding a figure.

Thank you, and in response, we added a new figure showing the main finding observed from the LID mouse to page 19.  

  • The results of the study demonstrated that while both groups of rats on a high-fat diet had similar weight gain, the GH/IGF-1 deficient Lewis Dwarf rats had elevated blood glucose levels, reduced insulin levels, and impaired glucose tolerance. Analysis of serum cytokine expression and aortic tissue revealed that GH/IGF-1 deficiency exacerbated inflammation, endothelial dysfunction, and oxidative stress in the GH/IGF-1 deficient Lewis Dwarf rats. This is an interesting link in that the loss of the GH/IGF-1 axis has led to heightened inflammation, do the authors know by what mechanism this happens?

Unfortunately, a precise mechanism is unknown, and we have added a paragraph between page 19, line 17- page 20, and line 7 to clarify.

  • In addition, previous studies using several transgenic mouse models, have demonstrated the crucial role of the GH-IGF-1 axis in the regulation of adipocyte proliferation and differentiation. The authors could briefly mention the process of adipocyte differentiation.

In response, we have added a paragraph between page 20, lines 15-20

  • Chapter 2 could benefit from a table to outline the key findings about the relationship between the GH and IGF-1 axis with obesity.

We appreciate your suggestion and have created a table that highlights the key findings regarding the relationship between the GH and IGF-1 axis in obesity on page 21.

  • Given that these terms were used previously in an earlier chapter, this level of definition and detail here might be a little out of place but ultimately this is your manuscript. Perhaps these could be moved up to chapter 1 and then the following chapters could then deal with primary studies. The logical flow of the topics is not great at present.

We agree and have kept most of the definitions in the earlier chapter but left a few for clarity as needed.

  • Figure 1 is useful but could also incorporate STAT5. Again, the IGF1 and GH axis has been referred to in an earlier section (I think it was the introduction), so maybe one solid section on this could be provided and then that could get referred to in any subsequent mention (to consolidate these two sections to prevent repetition).

Thank you.  A new figure was added to page 16 and referenced subsequently.

  • The primary study that follows then outlines the mechanism of the negative feedback of IGF-1 on GH via binding to its promoter. Another study using the MtT/S somatotroph cell line investigated the role of CREB-binding protein (CBP), required for the activation and regulation of GH gene expression, as a target of IGF-1 somatotroph regulation [82]. CREB binding by CBP requires the phosphorylation of CBP. In this regard, I am unsure what “required for the activation and regulation of GH gene expression, as a target of IGF-1 somatotroph regulation” means. I am aware that IGF1R can translocate to the nucleus, so what is binding the promoter of GH to regulate it? Is it P-CBP or IGF1R or some specific TF and co-actiavtor? Kindly make this clearer. Perhaps you could depict this in a figure.

To clarify the mechanism of negative feedback of IGF-1 on GH, we have added a new paragraph in the section on page 17, lines 6-19

  • Another comment about the logical flow of this manuscript. The mouse model of the authors gets referred to multiple times and it might be better to consolidate this information. I have also looked up your previous review (https://doi.org/10.3390/cells10102664) and the mouse model figure you have provided there is really informative and clear, perhaps something similar could be provided here so the authors can better understand the outcome of IGF-1R deletion (increased GHRH, GH, and IGF-1).

Thank you, we agree and have revised the manuscript to eliminate any repetition of information about our mouse model. In addition, we add a new figure to page 28.

  • Despite the established role of GH in promoting lipolysis, this mouse model study did not demonstrate any effect on total fat mass despite the significant elevation in circulating GH levels [20]. The authors could explain why this is the case and whether a parallel mechanism might be kicking in.

Thank you for your feedback. The lack of effect on total fat mass despite elevated GH levels is indeed intriguing. It is possible that compensatory mechanisms, such as changes in lipid metabolism or alterations in other hormones, are counteracting the impact of increased GH levels on fat mass. However, the authors of this paper did not examine differences in energy expenditure and glucose homeostasis between the control and GHRHΔIGF1R mice, making it difficult to speculate on the underlying mechanisms.

  • Recently, our laboratory has developed a new mouse model, referred to as the S- GIGFRKO mouse, which is characterized by the ablation of the IGF-1R in both the somatotrophs and GHRHexpressing neurons. The results of our study revealed that the SGIGFRKO mouse line displayed a modest increase in circulating GH levels and circulating IGF-1 levels. Does this suggest that a negative regulatory input from knocking out IGF-1R in somatostatin cells? The authors could explain this more in the text.

In the mouse model with a deletion of the IGF-1R from both somatotrophs and GHRH neurons, we observed an elevation in GH, and an increase in SST levels, which is likely compensatory.

  • The topics in section 5 (various mouse models 5.1- 5-9) could be summarised in a table. This will allow easier cross-comparisons between the models for any future reader. This way the readers can more easily choose their model of choice for their specific experiment.

Thank you for the suggestion. We have added a new table to the revised manuscript that summarizes the key features and characteristics of each mouse model discussed on pages 29-30.

  • Section 6. For this section, I would also suggest providing a figure about the contribution of these other hormones to the axis of GH/IGF-1.

Thank you, we agree that a figure demonstrating the contribution of other hormones to the GH/IGF-1 axis would be an excellent addition to the manuscript. A new figure was added to page 18

  • I appreciate that the authors have added information about the clinical trials but the abstract has outlined that detail about the management of obesity from the viewpoint of the axis of GH/IGF-1 will be provided, is there any other type of information about management (other than clinical trials) that can be added?

Thank you for your feedback. In the revised manuscript, we have added information about the management of obesity on page 4, line 20

We appreciate your feedback and guidance in improving our manuscript. Thank you again for your time and effort.

                 Sincerely,

Sarmed Al-Samerria and Sally Radovick

Reviewer 2 Report

The review article entitled "Exploring the Therapeutic Potential of Targeting GH and IGF-1 in the Management of Obesity: Insights from the Interplay between These Hormones and Metabolism" (ijms-2396451), is submitted to Molecular Endocrinology and Metabolism, in the Special Issue "Advances in the Understanding of Adipose Tissue Biology and Energy Metabolism".

It deals with a topic of great importance such as the situation of the high prevalence of childhood and juvenile obesity in the population with a growing trend. Its aim is to evaluate the current evidence on the therapeutic potential of targeting GH and IGF-1 in the management of obesity.

Comments:

The abstract, anticipates the information of the work therefore it should not only have the justification and the objective but also the methodology used and the main results. Currently, given the increasing research and scientific publication, review articles provide a complete view of the knowledge of the topics, but we see that there is an overlapping of reviews which leads to confusion. Therefore, I believe that the abstract should state the type of review conducted, the period being reviewed, and the databases used in this review as a minimum.

The introduction is very well presented although I suggest that the objective should not be divided into 3 sentences but should be more clearly stated, perhaps the main objective and secondary objectives could be used.

I think that the conclusions should be drawn from the review, I do not think that the safety of the use of GH and IGF-1 in the management of obesity is clear, so we should be more cautious in this regard.

As I have argued above, I believe that the review requires a section on methodology, which is currently fundamental, not only for the future but also to connect with other reviews that have been carried out.

Author Response

Dear Reviewer (2),

Thank you for your time and effort in reviewing our manuscript titled "Exploring the Therapeutic Potential of Targeting GH and IGF-1 in the Management of Obesity: Insights from the Interplay between These Hormones and Metabolism." We appreciate your thoughtful and constructive comments, which have helped us improve the quality of our review.

Comments:

  • The abstract anticipates the information of the work therefore it should not only have the justification and the objective but also the methodology used and the main results. Currently, given the increasing research and scientific publication, review articles provide a complete view of the knowledge of the topics, but we see that there is an overlapping of reviews which leads to confusion. Therefore, I believe that the abstract should state the type of review conducted, the period being reviewed, and the databases used in this review as a minimum.

Thank you for noting this oversight. We have revised the abstract to include information on the type of review conducted, the period being reviewed, and the databases used in the review on page 3.  

  • "The introduction is very well presented although I suggest that the objective should not be divided into 3 sentences but should be more clearly stated, perhaps the main objective and secondary objectives could be used"

Thank you. We revised our manuscript to ensure that our objectives are clearly stated on page 5 lines 18-27

  • "The article provides a comprehensive overview of the current evidence on the therapeutic potential of targeting GH and IGF-1 in the management of obesity. The use of a novel transgenic mouse model with a cell-specific deficiency of IGF-1R in the GHRH neurons of the hypothalamus and the pituitary somatotroph cells is an innovative approach, and the results are intriguing. However, it is important to note that animal models may not necessarily be directly applicable to humans, and more clinical trials are needed to fully evaluate the potential benefits and risks of GH therapy in managing obesity."

We agree entirely that animal models may not always directly translate to humans, and further clinical trials are needed to fully evaluate the potential benefits and risks of GH therapy in managing obesity. The new statement was added to page 6 lines 5-13 and page 27 lines 21-22.

  • I think that conclusions should be drawn from the review, I do not think that the safety of the use of GH and IGF-1 in the management of obesity is clear, so we should be more cautious in this regard.

Thank you for your feedback. After carefully considering your comments, we agree that caution should be exercised when considering the use of GH and IGF-1 in managing obesity. The conclusion has been updated to page 31.

  • “As I have argued above, I believe that the review requires a section on methodology, which is currently fundamental, not only for the future but also to connect with other reviews that have been carried out.”

Indeed, a new section was added to the MS pages 6-7.

We hope these revisions have addressed your concerns and improved the clarity of our manuscript. We appreciate your feedback and guidance in improving the quality of our manuscript.

Thank you again for your time and effort in reviewing our manuscript.

                 Sincerely,

Sarmed Al-Samerria and Sally Radovick

Round 2

Reviewer 1 Report

The authors have addressed my comments.

Reviewer 2 Report

Thank you very much for allowing me to review the new version of the manuscript entitled "Exploring the Therapeutic Potential of Targeting GH and IGF-1 in the Management of Obesity: Insights from the Interplay between These Hormones and Metabolism" (ijms-2396451). And the authors' response to the comments made.

All the suggestions made have been incorporated into the manuscript, and I have also identified improvements in the text that clarify important concepts, which undoubtedly involved other reviewers. The article is truly very interesting.

It is indeed a very interesting article that raises the possibility that GH and IGF-1 may offer a promising avenue for the management of obesity.

Minor comment: I suggest that the article's title indicates that it is a review.